# Optimization of Instrument Design for In-Line Monitoring of Dry Matter Content in Single Potatoes by NIR Interaction Spectroscopy

**DOI:** 10.3390/foods10040828

**Published:** 2021-04-11

**Authors:** Jens Petter Wold, Marion O’Farrell, Petter Vejle Andersen, Jon Tschudi

**Affiliations:** 1Nofima—Norwegian Institute for Food, Fisheries and Aquaculture Research, Muninbakken 9-13, Breivika, 9291 Tromsø, Norway; petter.andersen@nofima.no; 2SINTEF Digital, Smart Sensor Systems, Forskningsveien 1, 0373 Oslo, Norway; marion.ofarrell@sintef.no (M.O.); jon.tschudi@sintef.no (J.T.)

**Keywords:** inline NIR, interaction, potato, dry matter, near-infrared spectroscopy, industrial, PAT, process control, optical design

## Abstract

Dry matter (DM) content is one of the most important quality features of potatoes. It defines the physical properties of the potatoes and determines what kind of product the potatoes can be used for. This paper presents the results obtained by a novel prototype NIR (near-infrared) instrument designed to measure DM content in single potatoes in process. The instrument is based on interaction measurements to measure deeper into the potatoes. It measures rapidly, up to 50 measurements per second, allowing several moving potatoes to be measured per second. The instrument also enables several interactance distances to be recorded for each measurement. The instrument was calibrated based on three different potato varieties and the calibration measurements were done in a process plant, making the calibration model suitable for in-line use. A good calibration for DM was obtained by partial least squares regression (*RMSECV* = 0.78% DM, R^2^ = 0.91). The instrument was tested in-line in the process plant and several batches of potatoes were monitored for the estimation of the DM distribution per batch. Accuracy of DM determination as function of measurement position on the potato was studied, and results indicate that NIR scans along the center part of the potatoes give slightly better results compared to scans taken on either side of the center. Small differences in optical measurement geometry influence the accuracy of the calibration models, underlining the importance of optimizing instrument design for successful measurements.

## 1. Introduction

Dry matter (DM) content is one of the most important quality features of potatoes. It defines physical properties of the potatoes and determines what kind of products the potatoes can be used for. The DM content is frequently used to determine both the cost of the raw material and the optimal tuning of critical processing parameters. In an industrial process, such as French fry production, there will always be variation in DM content between potatoes within a batch and between batches. If this variation is large and cannot be handled, it will introduce unwanted quality variation in the end product and may also lead to food loss.

Today, most processing plants adjust their processes based on dry matter content measured by simple gravimetric methods, which are based on the density of the potatoes [1,2]. These estimates are based on sub-samples, typically 5–10 kg, taken from a large batch of several tons. Considering the variation within batches, this means that process adjustments are based on a very uncertain estimate of the average dry matter content. Furthermore, the variation within a crop is not even considered. This variation is known to be significant, depending on factors such as growing conditions and choice of cultivar [3,4]. Cole [3] found that DM content could vary from 17–24% within the same batch. An instrument that can rapidly measure the DM of every potato, early in the process, could be used for process optimization in the potato industry by, for example, sorting potatoes into batches with known and more uniform quality, and these can then be processed in an optimal way to ensure stable quality and prevent food loss. It could also be used to sort out potatoes with too low dry matter and use these for alternative products.

Near-infrared spectroscopy (NIRS) is well suited for determination of dry matter in various foods and fundamental work has demonstrated that this is the case also for potatoes. Haase et al. [5] obtained a coefficient of determination (R^2^) of 0.98 for estimation of dry matter content of homogeneous potato pulp. Dull et al. [6] showed that NIRS in transmission mode on thin slices of potatoes gave a correlation of 0.97 with DM, but this correlation decreased with increasing thickness of the slices. Potatoes are heterogeneous with prominent internal gradients in DM [7]. DM content is generally lower in the inner part and increases radially towards the outer part. There can also be systematic gradients along the tubers, which can vary between varieties and differently shaped potatoes. These heterogeneities make it challenging to obtain optical sampling with NIRS that represents the average DM content in the potato. Representative measurements can be improved by deep penetrating NIR and scanning along the potatoes. Subedi and Walsh [8] showed that it is possible to obtain quite good calibrations for DM in intact potatoes (prediction error of 1.52%) using interaction NIRS measurements, but the model was built using DM reference measurements from only the outer region of the potato where the NIR measurement was performed, and not the average DM content of the whole potato. Chen et al. [9] used interactance spectral measurements to determine carbohydrate content in intact whole potatoes and obtained a standard error of prediction (SEP) of 0.98%. Furthermore, it has been shown that a rapid NIR interaction point measurement system of relatively low spectral resolution (15 channels in the 780–1050 nm region) could be used for predicting dry matter (R^2^ = 0.95) in whole unpeeled potatoes off-line [4]. Using this system, the measured light in intact potatoes was found to have interacted with the potato from just below the surface down to a depth of approximately 20 mm. On the other hand, when using a commercial in-line NIR interaction imaging system, lower prediction performance (R^2^ = 0.83) was obtained [4]. The poorer performance with the latter instrument can most likely be ascribed to shorter interactance distances, resulting in a smaller fraction of light measured from deeper in the potato. This stresses that optical sampling is key to success with NIRS when measuring on heterogeneous samples. Helgerud et al. [10] observed that significantly lower prediction errors for average DM content in intact potatoes was obtained when they were scanned in movement (*RMSECV* = 1.06%) compared to when they were measured in steady state (*RMSECV* = 1.23%). The scanning ensured that a larger part of the potato was sampled and more representative spectra were collected. The listed studies concluded that NIRS is a very promising method for in-line quantification of DM in intact potatoes, but that no commercial NIR instrument at the time was suitable for industrial measurements. Such an instrument would need to handle the high speed of the moving potatoes, measure without physical contact and ensure sufficiently deep optical sampling. Applications of deep penetrating NIR spectroscopy are already established in the food processing industry for in-line quality grading of crabs [11], meat [12] and chicken fillets [13]. However, the system used for these applications does not perform satisfactorily for whole potatoes due to too limited optical sampling depth [4]. Cortés et al. [14] and Porep et al. [15] point out that NIRS is reported in numerous studies to be a promising method for in-line food applications, however, such implementations are rarely accomplished. Furthermore, a recent review calls for more efforts by researchers and industry to bridge the gap between the potential of NIRS and its actual implementation in process analytical technology [16].

The main aim of this R&D work was to develop and evaluate a novel NIR point measurement instrument that was specifically designed for demanding in-line applications, one of which was DM content in moving potatoes in a process line. The benefits of this measurement system include high signal to noise ratio (SNR) and simultaneous measurement of several interactance distances, allowing more controlled studies of optical sampling. The same system was successfully used to monitor core temperature in sausages in a modern industrial steam oven for continuous cooking of sausages [17].

In this study, the instrument was tested for non-contact continuous monitoring of streams of single potatoes. A DM calibration model was built for measuring potatoes in a French fry process, after peeling and heating, and before cutting and further processing. Three different potato varieties were included in the calibration. The calibrated NIR instrument was finally tested in the processing factory to monitor the variation in DM in different batches of potatoes. Robustness and repeatability were investigated in relation to the measurement orientation of the potatoes. Depth of sampling for different interactance distances were studied to further understand the performance.

## 2. Materials and Methods

### 2.1. Materials

#### 2.1.1. Calibration Set

A total of 163 whole potatoes of three different varieties (63 Innovator, 50 Folva and 50 Oleva) were randomly sampled from the process line in a French fry production plant in Norway. These varieties are typical for French fry production and usually span slightly different DM concentrations. The different varieties were collected on three different days. Folva and Oleva potatoes were collected from one batch each, while the Innovator potatoes were picked from two different production batches, supposedly with low and high average DM contents. The potatoes were grabbed directly from the conveyor belt after heating and peeling, and held at a temperature of about 40 °C. During calibration measurements, the NIR instrument was used off-line. However, it was located close to the production line so that the potatoes were measured approximately 30 s after removal from the belt, thus maintaining their temperature and humidity. After the NIR measurement, each potato was packed in a plastic bag, to avoid further drying, and stored overnight at 4 °C before measuring the reference dry matter the following day.

#### 2.1.2. Test Set

Six months after the calibration trial, the NIR instrument was installed in the process line above the conveyor belt, at the same point where the calibration samples were collected. Then, 50 potatoes (Innovator) were randomly sampled from two different production batches. To obtain controlled measurements, each potato was manually passed under the instrument for a 1 s measurement. They were then packed in plastic bags and stored overnight at 4 °C before measuring the reference dry matter the following day.

#### 2.1.3. Process Monitoring

One year after instrument calibration, the NIR instrument was again installed in the process line above the conveyor belt. Eight baches of Innovator potatoes were monitored on a moving conveyor belt, over two days, to explore DM variations between and within batches.

#### 2.1.4. Study of Robustness

Fifty potatoes from two batches of Innovator were collected from a potato farm. They were measured (NIRS) with skin and at a temperature of about 20 °C. They were then stored overnight at 4 °C before measuring the reference dry matter the following day.

### 2.2. NIR System

The prototype NIR system was designed to measure in non-contact interaction mode [14]. An illumination module, with a halogen light source (50 W), was designed with a light chopper to allow for continual background correction and a means for switching between several projected measurement geometries in real time (Figure 1a). Each illuminating line is 2 mm × 26 mm and the interactance signals between each pair of lines were measured consecutively, from a single 2 mm × 8 mm field of view (FOV) centered between the pairs. The light travels from the illuminated regions through the potato, exiting again for detection in the FOV. This was synchronized with a customized spectrometer with sufficient SNR for typical belt speeds in the industry. The distances between the pairwise illuminated regions were approximately 10, 16, 22, 28 and 34 mm, termed Dist1 to Dist5, respectively. A single recording contained spectra from each of the five measurement distances. Typically, several recordings were averaged per measurement to achieve sufficient SNR, giving a total measurement time of 0.1 s and upwards. The spectra consisted of twenty evenly spaced wavelengths in the region 761–1081 nm. The working distance between the instrument and potatoes was approximately 20 cm, depending on the size of the potato, meaning that the instrument was not in physical contact with the sample.

### 2.3. Calibration Procedure

Each of the 163 potatoes in the calibration set was measured on one side, once by being held still and once by passing it under the instrument while recording spectra. The collection time per measurement was 2 s, resulting in 100 NIR spectra. The 100 spectra obtained from each measurement on the stationary potatoes were averaged to one mean spectrum that was used for calibration. For potatoes passing under the instrument, typically it took 2 s from just before the potato entered the measurement area to then exiting the measurement area. It was important to extract and use only spectra that were measured on the potato. Detection of these relevant spectra was done automatically by a selection algorithm with threshold criteria for the intensity, shape and spectral contrast. The optimal potato spectra had a different shape compared to measurements on the very edge of the potato, or measurements of the metal surface the potatoes were placed on. The accepted spectra from each moving potato (typically 50–80 spectra) were averaged and the mean spectrum was used for calibration. The detected spectra were linearized and transformed to absorption spectra by taking the logarithm of the inverse of the interaction spectrum (log10(1/I)), where I is the intensity of the detected light. To minimize variation in the spectra induced by light scattering and varying distance between instrument and potato, the absorption spectra were normalized by standard normal variate (SNV). For each absorption spectrum the mean value was subtracted, and the spectrum was then divided by the standard deviation of the spectrum [18]. Our experience is that SNV gives good quantitative modelling properties and that it is more stable under in-line conditions than, for instance, multiplicative scattering correction (MSC).

Calibration models were made by partial least squares regression (PLSR) [19]. Two calibration models were made for the potatoes, one for spectra collected from stationary potatoes and one for spectra from potatoes in movement. The optimal number of factors in the models was determined after cross validation by evaluation of the squared correlation (R^2^) between measured and estimated DM, and the root mean square error of cross validation (*RMSECV*) defined as
(1)$$RMSECV=\sqrt{\frac{1}{N}\overset{N}{\underset{n=1}{\sum }}{\left(y_{n}-{\hat{y}}_{n}\right)}^{2}}$$
where *N* is total number of samples, *ŷ_n_* is the predicted value, *y_n_* is the measured reference value and *n* denotes the samples from 1 to *N*. The final regression vectors for the calibrations were implemented in the instrument for in-line testing.

Multivariate regression was done with software The Unscrambler 9.8 (Camo, Oslo, Norway). All other data processing, including true time data collection, detection algorithms and spectral pre-processing, was done by the use of Matlab version R2018a (MathWorks, Natick, MA, USA).

### 2.4. Process Monitoring

After calibration, the NIR instrument was installed in the process line above the conveyor belt, at the same position where calibration samples were previously collected. The purpose of the in-line trial was twofold: (1) To test the calibration by collecting and measuring a controlled test set of 50 new potatoes, and (2) the online collection of NIR spectra from large amounts of potatoes from different batches. The conveyor belt with potatoes was 50 cm wide, while the NIR instrument had a measurement area of approximately 34 mm (the distance between the farthest illumination stripes). It was, therefore, impossible to monitor the entire volume of potatoes. However, approximately 10% of the potatoes per batch was measured, giving a good representation of the dry matter distribution in the batches. During batch scanning the potatoes appeared at random positions and orientations on the belt. Again, an algorithm determined in true time when a potato was measured by using threshold criteria for the intensity, shape and contrast of the spectra. Due to the belt speed during the tests, typically 3–4 spectra were collected from each measured potato and the average of these was used to estimate the DM content. This means that scanning time for each potato was in the range 0.06–0.10 s. Eight batches were scanned.

For each batch, specific gravity for all potatoes in a 10 kg sub-sample was measured using a standard method based on immersion in a range of salt solutions [3]. This method gives the percentage distribution of potatoes at different DM contents.

### 2.5. Study of Robustness

Since the DM content is unevenly distributed in the potatoes, and since potatoes vary in size and shape, it is of interest to know if some measurement regions on the potatoes give better results than others. It was also of interest to know if the NIR measurements collected on the side of the potatoes were as useful when the potato was more sloped than the flatter part on the top/middle of the potato. Each of the 50 potatoes used in this study was scanned 12 times on a conveyor belt in a lab location. Two sides of each potato were scanned six times as indicated in Figure 1b, two times along the center line of the potato and four times closer to the side of the potato. The speed of the belt for these tests was slower than the one used in industry, so between 10 and 40 single spectra were collected per scan, meaning each scan took less that one second. Since the potatoes in this study had skin and were kept at 20 °C, it was not possible to use the established industrial calibration to estimate DM content so the study was focused on how the position of the scans affected estimated DM. To do this we made a calibration model based on the average of the 12 scans on each potato. This calibration was then used to estimate DM for each single scan to study the variation in DM predictions from measurements on the same potatoes. The calibration model was built in the same way as described above.

### 2.6. Study of Sampling Volume as Function of Distance between Illumination and Detection Region

To get an understanding of the extent of the measured volume for the different interactance distances, Dist1–Dist5, a peeled potato was cut so that one half had a starting thickness of 40 mm. It was then placed, flat side down, on a 4-cm-thick block of coconut fat. The fat was chosen because it gives a sufficiently different spectrum from potato with a distinct peak at 930 nm that is easy to distinguish from pure potato. The curved part of the potato faced the NIR instrument above. For each iteration, the potato was then sliced below, making it gradually thinner and thinner (a slice of approximately 1 mm was removed each time), and after each slice was removed the potato half was again placed on the coconut fat and an NIR spectrum was recorded. Acquisition time per measurement was 1 s. The same experiment was conducted with a potato with skin. Multivariate curve resolution (MCR) was used to extract the fat signal from the spectra [20], making it possible to estimate the relative contribution from fat as function of potato slice thickness, for each interactance distance. MCR was performed in Matlab version R2007b by the PLS_Toolbox (Eigenvector Research Inc., Manson, WA, USA).

### 2.7. Determination of Dry Matter

Each individual potato tuber was grated into strips (cross section of 2–3 mm) with a bench top grater (Hallde RG100, AB Hällde Maskiner, Sweden). Three sub-samples (20 g) of each potato tuber were immediately placed in aluminum pans and dried for 48 h at 105 °C in a forced fan oven. Dry matter percentages were calculated based on the weight before and after drying. Standard error of reference (*SER*) was calculated using the three sub-samples from each potato with the following formula:(2)$$SER=\sqrt{\left(\frac{\sum_{j=1}^{M}\sum_{i=1}^{N}{\left(Y_{ij}-{\bar{Y}}_{j}\right)}^{2}}{M\left(N-1\right)}\right)}$$
where *M* is total number of samples, *N* is number of sub-samples for each sample, *Y* is sub-sample DM and $\bar{Y}$ is average DM for the sample.

## 3. Results

### 3.1. Dry Matter Contents

All statistics for DM are summarized in Table 1. DM content in the calibration data set had a quite even distribution of DM over the whole range. The wide span of DM was a good basis for making calibration models, but also illustrates the challenge of optimizing industrial processing, since potatoes with a DM level of, e.g., 18%, would require different process settings than those with a level of 26%.

The test set of 50 randomly collected potatoes from the industrial line also revealed a large span in DM content. These potatoes were taken from the same production batch, which again illustrates the large internal batch variation. The last set of 50 potato samples, used for the robustness test, showed similar variation in DM as the test set.

Based on all the samples analyzed in the study, a SER of 0.40% DM was calculated.

### 3.2. Spectroscopic Measurements

Typical absorption spectra from a potato are shown in Figure 2a. The five spectra are from the five different distances from the two illumination stripes to the FOV. The absorption increases systematically with increasing distance since the light travels farther before it is detected. The quantitative difference in absorption between Dist1 and Dist5 corresponds to an approximately 24 times higher signal intensity for signals from Dist1. Figure 2b shows the corresponding spectra where the mean value for each spectrum has been subtracted and this reveals that the contrast in the spectra increases systematically with increasing distance, meaning that spectral features are more pronounced when measured at longer distances. Note, however, that for Dist5 there is a smaller change in contrast at the water peak (seen at around 980 nm) versus Dist4 than what is seen between the other distances. This might be due to a very strong absorption due to the long travelling distance, giving non-linear changes in absorbance. The weaker signal intensity will also be more disturbed by stray light reflected directly off the surface.

Figure 3a shows spectra from potato slices of different thicknesses measured on a block of fat. The thinner the potato slice is, the more pronounced the fat peak at 930 nm is. Based on MCR it was possible to separate the signals from fat and potato, and the estimated pure spectra are shown in Figure 3b. The corresponding estimated concentrations of fat (as percent of a spectrum of pure fat) as function of potato slice thickness for the different distances are plotted in Figure 3c. They show that the fat is detectable down to depths of 10–11 mm with Dist4–5 while only to about 6 mm for Dist1. With a 5 mm potato slice, the proportion of fat in the spectrum is 2% for Dist1 compared to 15% for Dist5. The larger the distance, the more suppressed the higher-intensity light from the shallow layers in the potato is. For potatoes with skin (results not shown) we saw the same systematic differences with distance, but with generally smaller contributions from fat. This is likely due to the skin absorbing light and reducing the sensitivity to light from deeper in the potato. Related studies indicate a sampling depth of 4 mm in the 700–900 nm range in intact apples with NIR reflection [21] and 11–12 mm in fish cakes with NIR interaction [22]. The results illustrate that the distance between illumination and detection region largely affects how deep we are probing in the potatoes. Since potatoes are heterogeneous with internal gradients in DM concentration, it would most likely be favorable to probe as deep as possible as long as the signal has a sufficient SNR.

SNV-transformed spectra of the Innovator calibration potatoes are shown in Figure 4. They are dominated by the water absorption band around 980 nm. The colour code indicates some visible systematic variation as a function of DM content, but the regression models indicate that this apparent variation explained only 45% of the variation in DM. The regression coefficients (Figure 4b) can indicate which absorption bands contribute with quantitative information. The main contributions come from the regions 900–940 nm and 960–980 nm. The region around 960 nm is known to contain the second overtone from the O–H bond in water molecules, and is therefore negatively weighted. The C–H bond in starch has a strong absorption band at 979 nm and explains the positive contribution from this area. Furthermore, 878 nm and 901 nm are also pointed out as two starch absorption bands [23]. The regression coefficients are very similar to those obtained by Helgerud et al. [10].

### 3.3. Calibration Results

The calibration results are summarized in Table 2. Firstly, the calibration models were more accurate for potatoes that were scanned in motion versus those that were measured in steady state. This is reasonable since scanning allows for collection and averaging of spectra along the entire potato tuber, not just at the middle of the potato (see Figure 1) and, as previously discussed, DM is unevenly distributed in the potatoes [7,10]. Helgerud et al. [10] also obtained better results for potatoes in motion compared to those in steady state. Secondly, the prediction errors (*RMSECV*/RMSEP) were notably higher than the calculated SER. This is mainly due to the difference in sampling between NIR measurements and the reference method; the NIR system probed approximately the upper 6–10 mm layer of the potato, while the reference values were based on measuring the entire potato. This means that the calibrations relied on the correlation between DM in the outer part of the potato and the average DM content of the whole potato. This is not optimal, as pointed out by others [24], since this relation could vary with variety, season and soil type, but the target quality parameter is the average DM when processing potatoes, not the DM content in the outer part of the potatoes.

There were also clear systematic differences in model performance as a function of interactance distances, i.e., the optical measurement geometries Dist1–Dist5. For measurements in both steady state and movement, best results were obtained for Dist4. A plot of predicted versus measured DM in potatoes in motion is shown in Figure 5a. Shorter distances gave gradually and significantly higher prediction errors. As discussed above, the distance between field of illumination and detection affects depth of optical sampling, and this depth is important for optimization. The simultaneous collection of spectra from different distances on large data sets gives a unique ability to determine the optimal geometry through regression modelling. Helgerud et al. [10] used an NIR instrument with larger and less defined illumination stripes, and a fixed interaction distance between the field of view and the edge of the illumination. They obtained slightly higher prediction errors (*RMSECV* = 1.06%) than the optimal geometry in this study.

We also found that combining (concatenating) the spectra from two distances, e.g., Dist2 and Dist4, did not give better results (not shown here). Poorer accuracy was achieved when averaging the spectra from more than one distance, e.g., the average of Dist2, Dist3 and Dist4, or the average of Dist2 and Dist4 (also not shown here), indicating that mixing signals from different distances is less optimal even though the SNR increases.

The same variation in accuracy as function of distance was observed for the test set, verifying the results from the calibration analysis. The RMSEP values for the test set were slightly higher compared to the *RMSECV* for the calibration models. This is quite common and expected when the calibration is built from data measured in a slightly different environment than on the actual process line. The potatoes were also from other batches and stored a longer period before processing, which could introduce different chemical and physical properties affecting the NIR spectra. It is well known that calibration models based on NIR spectroscopy for apples are very sensitive to variety, growing location and season [25]. Similar studies have not been conducted for potatoes but should be done to clarify potential costs related to calibration and calibration maintenance. It has been observed that NIR measurements from different potato varieties tends to cluster in a principal component analysis [10], but since DM also varies systematically between varieties, contributing to similar clustering, this makes causality less straightforward to determine.

It must be noted that the R^2^ values were considerably lower for the test set than for the calibrations. This was due to small systematic differences in the spectra recorded on potatoes in-line compared to those recorded during calibration. This gave a slope of 0.65 and an offset of 7.5% for the predicted values (for Dist4). Before the calibration model was implemented for in-line collection of batch data it was *slope* and *offset* corrected according to Equation (2)
(3)$${\hat{y}}_{n_{corrected}}=\left({\hat{y}}_{n}-offset\right)/slope$$
where *ŷ_n_* is the predicted value from the original calibration model. Figure 5b shows the corrected prediction values of the test set. After correction the slope was 1, R^2^ increased to 0.81 while the RMSEP increased to 0.90%. A slight increase in prediction error per potato was acceptable. The alternative would be that potatoes with high DM content would be systematically underestimated while potatoes with low DM content would be overestimated. The average DM values from several potatoes would be more on target with the slope corrected calibration.

### 3.4. Process Monitoring

The number of potatoes measured in the scanned batches varied from 1742 to 4861 with 3500 as an average number. The batches were measured over two days and the variation between the batches was not large. Figure 6 shows the measured dry matter distribution in four of the batches, where Batch 1 had the highest average DM content (23.7%) while Batch 3 had the lowest (21.7%). The histograms reveal the distribution of DM, which is also useful to know when processing the potatoes. DM distribution is routinely checked using lab measurements based on 10 kg of potatoes from each batch of several tons. These standard measurements matched the in-line measurements quite well, except for Batch 1, which was slightly skewed toward lower values. With test samples representing less than 1% of the total batch, there is always a chance that the sample is not representative for the whole batch, which can lead to sub-optimal processing. Regardless, the test sample distributions in this case did to some extent verify that the NIR instrument produced reasonable results.

A main uncertainty with the in-line prediction values is that each mean spectrum from the potatoes was based on 3–5 single spectra, fewer than in the calibration and test sets. This could result in more noisy and less representative spectra, giving higher prediction errors per potato. A more detailed study of how speed affects the prediction error would therefore be of value. However, while the accuracy per potato may be lower, with the high number of potatoes measured per batch, it can be assumed that the accuracies of the average value and distribution per batch are high. This was previously shown for in-line measurements of fat in batches of beef, where the prediction error per batch was reduced significantly with increasing size of the batch [12].

### 3.5. Study of Robustness

As described above, a study of robustness was conducted on 50 potatoes. Table 3 summarizes calibration and test results for the 50 potatoes that were scanned 12 times each. The estimated prediction errors (*RMSECV*) for the calibrations were considerably lower than for the calibrations made for industrial use. The main reason for this was that the spectra used in these models were the average of the 12 scans per potato, which was then more representative of the average DM content in the potato than just one single scan, as was used in the industrial calibrations.

Another interesting result was that the optimal distance between the two illuminated lines was different for potatoes with skin versus heated and peeled potatoes. In this case, Dist2 and Dist3 gave better models than Dist4. The main reason is most likely that the skin absorbs much of the light, so that the light from the longer Dist4 was weaker, with lower SNR. The industrial heating process could also induce other optical properties in the potatoes, which could explain this difference. It should be noted that the averaging of 12 spectra is an unlikely explanation of this difference, since Dist2 and 3 were better even when models were based on single scans from the potatoes with skin (results not shown). These results again underline that small design details in instrument and geometry are important for robust performance.

Prediction errors based on prediction values from the single scans were more on par with those in Table 1. Slightly more accurate predictions were again obtained when the potatoes were scanned along the center line, versus towards the sides. There can be at least two explanations for this. (1) Scans along the center line are more representative of the average DM content in the potatoes since DM is concentrically distributed with high concentrations in the outer layers and lower concentrations in the inner layers. An NIR measurement that is perpendicular to the center of the potato will likely probe this concentric distribution better than a measurement on the side of the potato. (2) NIR signals measured on the side of the potatoes, where the surface being measured is more sloped, are geometrically less well defined and are, therefore, distorted by unequal measurement conditions on each side of the field of view. The results indicate that it would be an advantage that the potatoes are positioned on the belt so that the center line of the potatoes are scanned, which can be achieved by mechanical guides.

## 4. Conclusions

The results in this study indicate that the presented NIR instrument has the performance needed for successful use in potato processing. Specifically:It can probe light that has travelled deeply enough into the potatoes, by suppressing the stronger light closer to the surface. This in turn enables sufficiently good DM estimates due to better sample representation. It has earlier been reported that depth of measurement is critical for this application [4].It produces spectra with sufficient signal to noise ratios, enabling robust calibration models for DM, with acceptable accuracy.Measurements are done without physical contact, with a distance of approximately 20 cm between the instrument and potatoes.It can operate at high speed, capturing NIR spectra from single potatoes within 0.02 s.

The results also illustrate that accuracy of the calibration models is significantly dependent on the optical measurement geometry. Such instrument optimization is crucial to succeed with NIRS-based measurements of complex samples in process lines. This study documents the feasibility, but it is recommended that an evaluation of the calibration robustness over a longer period, and on different varieties of potatoes, should be conducted. Different potato varieties might need individual calibration models due to differences in internal physiology. The system here is evaluated for whole potatoes; it could work well also for DM quantification in potato strips in a French fry process. For this application, it could be of interest to monitor both DM and fat at certain steps in the process to optimize drying time and frying time to obtain high process yield and a stable end quality.

## Figures and Tables

**Figure 1 foods-10-00828-f001:**
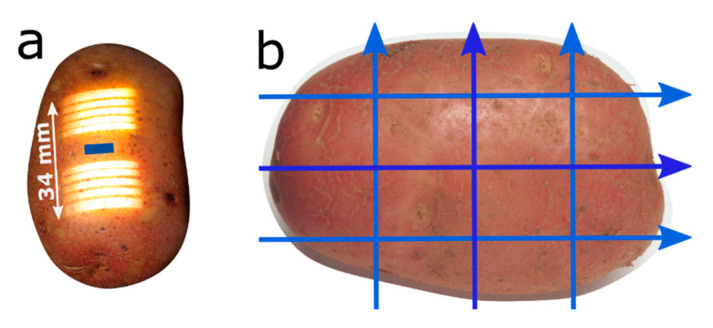
(**a**) Illustration of illumination pattern and the field of view of the system (blue rectangle). (**b**) Illustration of position of the FOV (field of view) during the scans during the robustness study. This pattern was used on two sides of each potato, center scans and off-center scans.

**Figure 2 foods-10-00828-f002:**
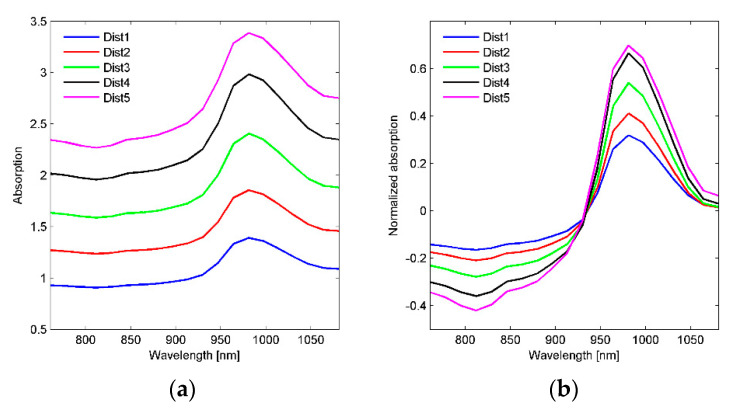
(**a**) Absorption spectra for the five different distances from one Innovator potato, indicated with Dist1 to Dist5. (**b**) Same measurements as in 2 (**a**), but with mean value subtracted from each spectrum.

**Figure 3 foods-10-00828-f003:**
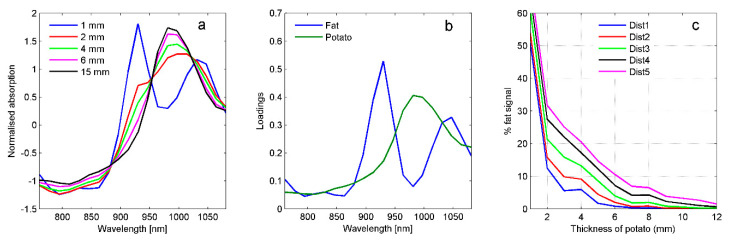
(**a**) SNV (standard normal variate)-corrected absorption spectra from potato slices of different thicknesses on a sample of coconut fat (shown for Dist4). (**b**) Pure spectra of fat and potato as estimated by MCR (multivariate curve resolution). (**c**) Estimated concentrations of fat as function of potato slice thickness for Dist1–Dist5.

**Figure 4 foods-10-00828-f004:**
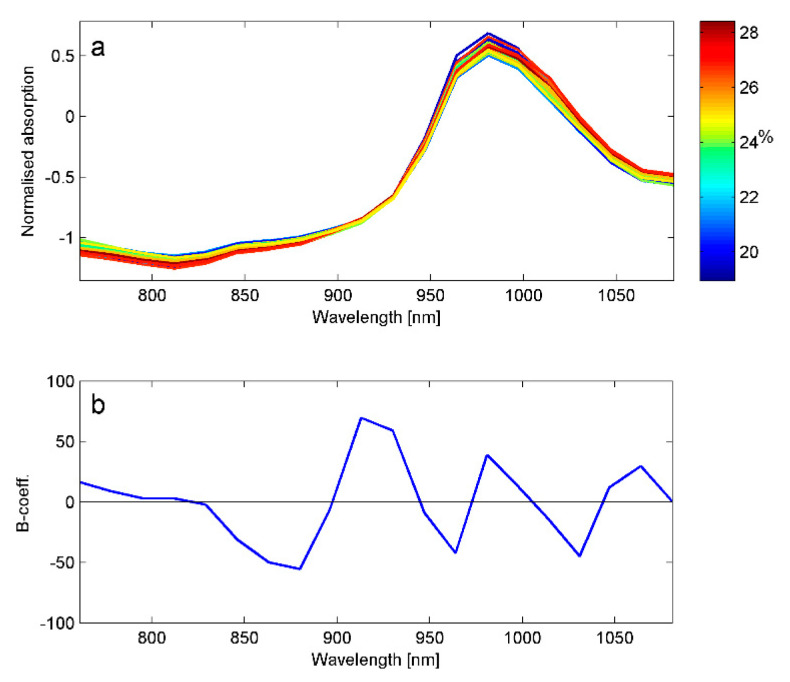
(**a**) SNV-corrected spectra from Innovator potatoes colored according to dry matter content. (**b**) Regression coefficients for the calibration model for dry matter in potatoes.

**Figure 5 foods-10-00828-f005:**
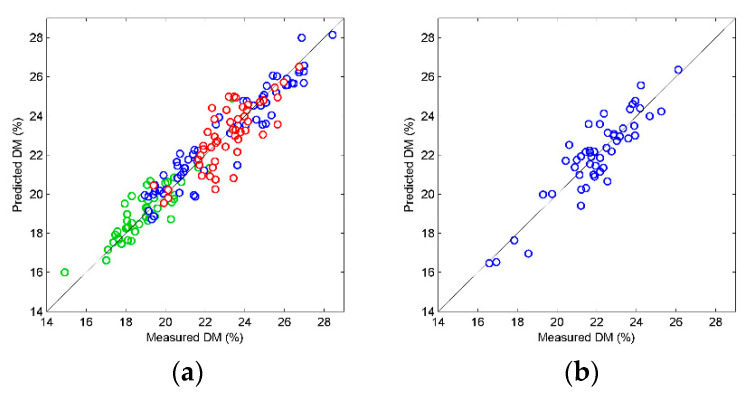
(**a**) Predicted versus measured dry matter content for cross validated calibration model. Colors refer to the different potato varieties: Innovator (blue), Oleva (green), Folva (red). (**b**) Predicted dry matter in test set measured on industrial conveyor belt. Values are slope and offset corrected.

**Figure 6 foods-10-00828-f006:**
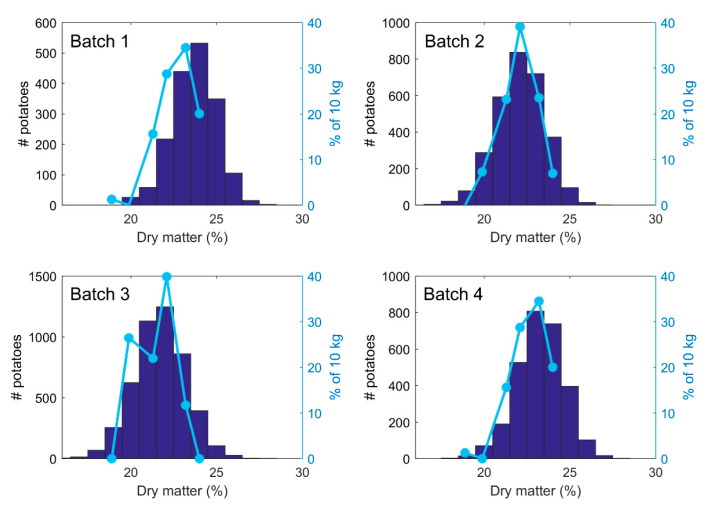
NIR estimated dry matter in four batches of potatoes summarized in histograms. Blue lines indicate distribution of dry matter measured on a 10 kg subsample of the batch.

**Table 1 foods-10-00828-t001:** Summary of statistics for reference measurements of DM (dry matter) in potatoes. All values are given in % DM.

Variety	N ^a^	Min	Max	Average	StDev ^b^
Oleva (calibration)	50	19.4	26.7	23.1	1.53
Folva (calibration)	50	14.9	23.4	19.1	1.48
Innovator (calibration)	63	18.9	28.4	23.0	2.68
All calibration samples	163	14.9	28.4	21.8	2.75
Test set	50	16.6	26.1	22.0	1.91
Robustness test	50	17.8	25.7	22.9	1.72

^a^ Number of samples, ^b^ Standard deviation.

**Table 2 foods-10-00828-t002:** Calibration results for potatoes measured in steady state and in motion for different distances between illuminated region and area of detection. Results for test samples are based on calibrations for potatoes in motion.

	Steady State			Scanning in Motion			Test In-Line	
Dist	#LV ^a^	R^2^	*RMSECV*^b^ (%)	#LV ^a^	R^2^	*RMSECV*^b^ (%)	R^2^	RMSEP ^b^ (%)
1	7	0.76	1.35	5	0.83	1.11	0.52	1.30
2	7	0.79	1.25	7	0.87	0.91	0.72	1.00
3	5	0.81	1.17	7	0.89	0.89	0.74	0.94
4	5	0.84	1.10	7	0.91	0.78	0.77	0.88
5	5	0.82	1.14	7	0.87	0.98	0.63	1.14

^a^ Number of latent variables used in model. ^b^ Root mean square error of cross validation/prediction.

**Table 3 foods-10-00828-t003:** Calibration results for potatoes with skin measured in motion for Dist1–Dist5. Prediction errors (RMSEP) are listed for center, off-center and all scans for the different distances.

	Calibration Set			RMSEP ^b^ (%)		
Dist	#LV ^a^	R^2^	*RMSECV*^b^ (%)	Center	All	Off-Center
1	6	0.85	0.65	0.74	0.86	0.89
2	6	0.90	0.53	0.70	0.79	0.83
3	5	0.90	0.51	0.83	0.98	1.04
4	5	0.85	0.65	1.26	1.35	1.44
5	4	0.81	0.74	1.49	1.61	1.66

^a^ Number of latent variables used in model. ^b^ Root mean square error of cross validation/prediction.

## Data Availability

The data presented in this study are available on request from the corresponding author.

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
