# Peer review of "Optimization of Instrument Design for In-Line Monitoring of Dry Matter Content in Single Potatoes by NIR Interaction Spectroscopy"

_foods, 2021, doi:10.3390/foods10040828_

Round 1

Reviewer 1 Report

Dear authors, 

I have found the submitted Manuscript as one of the really well-written, scientifically sound, with all the necessary information for replication of measurements, and very clearly explained results, which support the conclusions. The experience you share in this paper can be very useful not only for PAT in food control, but many other applications. Hence,I would recommend revise minor grammar mistakes. My comments on what to correct can be found below, and regarding the questions I asked, I do not think it would be necessary to answer (and revise the paper to include it), I think it would benefit those readers interested to understand in more depth the phenomena you utilized for practical purpose. But as I said, this is optional. 

The comments and questions: 

L186 SNV - did you try other types of preprocessing? SNV seems to me to distort a little bit the spectra on the edges of the region. This can result in shifting of absorbance bands and misleading information about the most influential variables. Maybe some locally weighted preprocessing would result in lower SEP and better understanding of the needed instrumental setup. For example of SNV related issues see: 

Oliveri, Paolo, et al. "The impact of signal pre-processing on the final interpretation of analytical outcomes–A tutorial." Analytica chimica acta 1058 (2019): 9-17.

L210 mm - please add space between numbers and units

L222 per centage - please correct this to one word

L266 Y_  - please write the average correctly, with line above the Y

L334 results illustrates - correct to results illustrate

L351 models indicates - same as the above (plural noun, no "s" in verb)

L352 I don't agree with the wording "can often indicate". It is not "often", simply this is not the matter of frequency - the coefficients CAN provide this information.  

L355-357: C-H bond at 979 nm. 878 nm and 901 nm - starch - the signal is dominated by water, and it seems by strongly bound water (to dry matter). I think the interpretation might be different, especially for the 979 nm band. I don't think you look at dry matter, you look at the water hydrating that dry matter. Maybe this is worth exploring. 

L380-381 - there is always a part of water that can not be removed by drying. This also contributes to error, it is not just because of the sampling, but as I think you say later in the text, simply the reference method measures a bit different thing. Your results might actually be more accurate. Good thing to check is to do Principal Component Regression and check how it compares with PLSR. PLSR is influenced by the errors of reference method. 

L436 correct yn 

Figure 5 - why do you aim to develop a general model? Different variety doesn't mean only different DM content, but many more chemical and physical attributes. Why not local model for each variety? Wouldn't that be more accurate? 

505-heating drastically influences the water structure, and especially in food samples, like your potatoes, the water which is the "glue" for all other components, is connected with many macro-properties of food, such as texture, hardness/softness, shelf-life etc. The structure of water governs those features. The most influential factor on the structure of water is temperature. You might want to look for example studies such as this: 

Vanoli, Maristella, et al. "Water spectral pattern as a marker for studying apple sensory texture." Advances in Horticultural Science 32.3 (2018): 343-351.

I think from the data you have, you can see much more about the potatoes, not only one, single variable DM. 

The lines at which the grammatical corrections would be needed are highlighted in the attached pdf. 

Author Response

Dear Referee,

First of all, thank you for a positive review of our paper. I have answered your comment below, in red text.

L186 SNV - did you try other types of preprocessing? SNV seems to me to distort a little bit the spectra on the edges of the region. This can result in shifting of absorbance bands and misleading information about the most influential variables. Maybe some locally weighted preprocessing would result in lower SEP and better understanding of the needed instrumental setup. For example of SNV related issues see: 

Oliveri, Paolo, et al. "The impact of signal pre-processing on the final interpretation of analytical outcomes–A tutorial." Analytica chimica acta 1058 (2019): 9-17.

Thank you for the suggested article by Olivieri et al. Why SNV? After some years of working with interactance NIR in inline situations we have landed on SNV as the most stable pre-processing. The conclusion is based more on empirical experience than on theoretical studies. This is why we do not give a clear explanation for our choice. For these kind of low resolution spectral data, derivatives do not make much sense. MSC and EMSC are interesting approaches that we have tested, but there are two drawbacks compared to SNV:

  1. They tend to give slightly less accurate model performance with our data.
  2. Model performance in industrial in-line situation is often much more unstable (in particular for EMSC). This is because we rely on the (E)MSC model for pre-processing, and we have seen that these models can give pre-processing of spectra that results in way off prediction values. So, when using MSC, it is important to have robust MSC models that are representative for all new samples. Our experience is that SNV is more stable in in-line situations and requires less work.
  3. I am aware of that SNV can distort spectra with regard to interpretation. The data we use here covers a rather narrow spectral range (780-1080 nm) where there are few prominent peaks. Then the spectral shapes are quite well maintained after SNV and sound interpretation is still possible to do. Personally I often use just raw absorption spectra + PCA for interpretation. In this case, when main aim is quantitative regression, and not interpretation, then SNV is fine.
  4. One sentence of explanation is added in the paper.

L210 mm - please add space between numbers and units Done

L222 per centage - please correct this to one word Done

L266 Y_  - please write the average correctly, with line above the Y Done

L334 results illustrates - correct to results illustrate  Done

L351 models indicates - same as the above (plural noun, no "s" in verb) Done

L352 I don't agree with the wording "can often indicate". It is not "often", simply this is not the matter of frequency - the coefficients CAN provide this information.  Agree. Change made.

L355-357: C-H bond at 979 nm. 878 nm and 901 nm - starch - the signal is dominated by water, and it seems by strongly bound water (to dry matter). I think the interpretation might be different, especially for the 979 nm band. I don't think you look at dry matter, you look at the water hydrating that dry matter. Maybe this is worth exploring. Thank you for suggestion. I think we leave the interpretation as is now, and we might do additional experiments to investigate this further. I agree that it is important and also very satisfactory when a spectral interpretation is most likely correct. In this case it is slightly speculative, but we do pinpoint the absorption regions, which are the most likely into play.   

L380-381 - there is always a part of water that can not be removed by drying. This also contributes to error, it is not just because of the sampling, but as I think you say later in the text, simply the reference method measures a bit different thing. Your results might actually be more accurate. Good thing to check is to do Principal Component Regression and check how it compares with PLSR. PLSR is influenced by the errors of reference method. I will se if PCR gives different results. But I think in his case, the sampling error/mismatch between spectroscopy and the reference method is the major source of error.

L436 correct yn done

Figure 5 - why do you aim to develop a general model? Different variety doesn't mean only different DM content, but many more chemical and physical attributes. Why not local model for each variety? Wouldn't that be more accurate?  If it is possible to use a general model that works well enough, it would be practical. I agree that it would most likely be more optimal to make models for different varieties, and this might be the way to go for a future implementation. The spread in DM for two of the varieties in this study was rather short to give stable models. The only variety in this study that had a sufficient variation was Innovator, and a model based on this one only got slightly lower prediction errors. A practical approach could also be to use locally weighted regression, independent of variety. We might study this the nest years. I have added a sentence about the potential need of individual models for different potato varieties in the conclusion.

505-heating drastically influences the water structure, and especially in food samples, like your potatoes, the water which is the "glue" for all other components, is connected with many macro-properties of food, such as texture, hardness/softness, shelf-life etc. The structure of water governs those features. The most influential factor on the structure of water is temperature. You might want to look for example studies such as this: 

Vanoli, Maristella, et al. "Water spectral pattern as a marker for studying apple sensory texture." Advances in Horticultural Science 32.3 (2018): 343-351.

I think from the data you have, you can see much more about the potatoes, not only one, single variable DM. Thank you also for this suggested paper. The focus of our paper was DM, so we did not investigate e.g. texture. But I will read the paper for inspiration. I have done much work on meat and seafood where the state of water is of great importance (temperature, water binding to proteins etc) so I am aware of that water is not just water.

Reviewer 2 Report

The article "Optimization of instrument design for in-line monitoring of dry matter content in single potatoes by NIR interaction spec-troscopy" is interesting and of high value.

Overall, it looks like a well-written and well-organized paper

A new use of NIR instruments is presented and a new test to be used in an industry where no other analitical techniques are disponible.

Also conclusion are weel written and authors correctly highlits that only expert can use the NIR instrument and that a good calibration set is needed to use it

INTRODUCTION

Line 30-37: add references

M&M

I think that more batches are needed to set up a correct calibration set. Explain why do you use only one batch and underline in results/conclusion that results could be influenced by this data.

Did you try other types of preprocessing?

Did you try also a basic PCA analysis?

Are you sure that water don't influence analysis?

Temperature and time are monitored?

Minor comments:

L210 mm - please add space between numbers and units - percentage: one word

L334 results illustrates - correct to results illustrate L351 models indicates - correct to models indicate

Author Response

Dear Referee,

First of all, thank you for a positive review of our paper. I have answered your comment below, in red text.

Line 30-37: add references

All the needed references to this first paragraph are sited in the next paragraphs where we go in more detail. So there is no need for additional references here.

I think that more batches are needed to set up a correct calibration set. Explain why do you use only one batch and underline in results/conclusion that results could be influenced by this data.

Maybe the referee did not read the manuscript thoroughly? It is clearly stated that the calibration set consisted of potatoes from three different varieties and from four different batches. Furthermore, the model was tested on a dataset from two different batches. So the material for the model should be sufficient.

A calibration set is seldom corrector not correct. But it is important that it is representative for the samples that are to be estimated.

Did you try other types of preprocessing? Yes, we did also try other preprocessing. We selected SNV because it gave good results and also because it gives stable results under in-line conditions. A sentence has been added about this.

Did you try also a basic PCA analysis? PCA is something we always do to explore spectral data. But this analysis is not included in this paper. Here we wanted to establish a quantitative regression model, and that is not done with PCA.

Are you sure that water don't influence analysis? It depends on what you mean. Of course water influences the measurements. Potatoes contain 70-82% water and this is the main part of the NIR signal. But this is also part of the model. If you ask about the water on the surface of the potatoes, I think it has very little  influence since we are measuring in interactance and not reflectance.

Temperature and time are monitored? Not sure what you point at here? Temp. in potatoes was measured. Time of scanning was also monitored.

Minor comments:

L210 mm - please add space between numbers and units - percentage: one word done

L334 results illustrates - correct to results illustrate L351 models indicates - correct to models indicate done